# InfoNCE is variational inference in a recognition parameterised model

**Laurence Aitchison**                                          *laurence.aitchison@gmail.com*
*University of Bristol*

**Stoil Ganev**                                                 *stoil.ganev@bristol.ac.uk*
*University of Bristol*

Reviewed on OpenReview: *https://openreview.net/forum?id=chbRsWwjax*

## Abstract

Here, we develop a new class of Bayesian latent variable model, the recognition parameterised model (RPM). RPMs have an implicit likelihood, which is defined in terms of the recognition model. Therefore, it is not possible to do traditional "generation" with RPMs. Instead, RPMs are designed to learn good latent representations of data (in modern parlance, they solve a self-supervised learning task). Indeed, the RPM implicit likelihood is specifically designed so that it drops out of the VI objective, the ELBO. That allows us to learn an RPM without a "reconstruction" step, which is believed to be at the root of poor latent representations learned by VAEs. Indeed, in a very specific setting where we learn the optimal prior, the RPM ELBO becomes equal to the mutual information (MI; up to a constant), establishing a connection to pre-existing self-supervised learning methods such as InfoNCE.

## 1 Introduction

Self-supervised learning (SSL) involves learning structured, high-level representations of data useful for downstream tasks such as few-shot classification. One common self-supervised approach is to define a "pretext" classification task (Dosovitskiy et al., 2015; Noroozi & Favaro, 2016; Doersch et al., 2015; Gidaris et al., 2018). For instance, we might take a number of images, rotate them, and then ask the model to determine the rotation applied (Gidaris et al., 2018). The rotation can be identified by looking at the objects in the image (e.g. grass is typically on the bottom of the image, while birds are nearer the top), and thus a representation useful for determining the orientation may also extract useful information for other high-level tasks. We are interested in an alternative class of SSL objectives known as InfoNCE (NCE standing for noise contrastive estimation; (Oord et al., 2018; Chen et al., 2020)). These methods take two inputs (e.g. two different frames of video), encode them to form two latent representations, and use a classification task to maximize a bound on the mutual information between latent representations. As the shared information should concern high-level properties such as objects, but not low-level details of each patch, this should again extract a useful representation, which is also invariant to data augmentation.

Ultimately, the goal of SSL is to extracting useful, structured, high-level representations of data. Interestingly, such representations have classically been obtained using a probabilistic latent variable models (PLVMs; Murphy, 2022). For instance, consider Latent Dirichlet Analysis (LDA; Blei et al., 2003). LDA clusters words and documents into topics. Given a new document, LDA allows us to assign a combination of topics to that document. Importantly, LDA, as a *probabilistic* latent variable model, allows us to sample "fake data". However, in the case of LDA, it is not at all clear why you would ever want to sample fake data. Specifically, LDA treats documents as "bags of words"; while you could sample bags of words, it is unclear why you ever would ever want to. Alternative classic PLVMs include probabilistic PCA/ICA (Pearlmutter & Parra, 1996; MacKay, 1996; Tipping & Bishop, 1999). Again, in probabilistic PCA/ICA, the point was

always to extract interpretable, high-level factors of variation. While you could sample "fake data", it is again not at all clear why you would ever want to.

Perhaps the most obvious modern incarnation of traditional PLVMs is the variational autoencoder (VAEs Kingma & Welling, 2013; Rezende et al., 2014; Kingma et al., 2019). The VAE extracts latent variables, and one would hope that these latents would form a good high-level representation, so that VAEs could be used in SSL. However, VAEs typically perform poorly when evaluated on SSL tasks such as few-shot classification or disentanglement (Karaletsos et al., 2015; Higgins et al., 2016; Zhao et al., 2019). The issues are commonly understood to arise because a VAE needs to reconstruct the data, so the latent representation is forced to encode low-level details (Chen et al., 2020; Balestriero & LeCun, 2024). This raises an important question: is it possible to develop new PLVMs that give useful high-level representations useful in SSL by eliminating the need to "reconstruct"?

Here, we answer in the affirmative by developing a new family of recognition parameterised model (RPM) (note that the "recognition parameterised model" name first appeared in follow-up work Walker et al., 2023b but the actual idea was introduced here; see Related work for further details). We perform variational inference in this RPM, so the resulting objective is the ELBO. Next, we connect the RPM to modern SSL methods, in particular InfoNCE (Oord et al., 2018; Chen et al., 2020). We show that the RPM ELBO is equal (up to a constant) to the MI under an optimized prior (Sec. 4.4), and equal (up to a constant) to the infinite-sample limit of the InfoNCE objective under a different choice of prior (Sec. 4.5). We give an example in a toy system with a latent space without unit-norm constraints in which the usual choice of InfoNCE objective fails completely, but a modified system that exploits our prior knowledge about dynamics in the latent space succeeds (Sec. 5).

This aligns with recent work (Locatello et al., 2019) which argues that good, problem-specific inductive biases are critical for effective disentanglement. However, it is difficult to introduce such inductive biases in traditional InfoNCE like settings. In contrast, it is straightforward to introduce these inductive biases into the priors of an RPM.

At a high-level, RPMs function by having observations that consist of multiple components, e.g. following a classic SSL setting, we could have two components, where each component is a different augmentation of the same underlying image. RPMs then use a latent space which models only dependencies between components, and avoids modelling any information about the marginal distribution over each component. Information about potentially complex marginals is instead captured in a complex likelihood. This likelihood is difficult to specify, and lies at the root of problems around "reconstruction" in classical VAEs. Critically, in the RPM framework the likelihood cancels out of the ELBO (see Sec. 4). In this way, an RPM is able to specify a full generative model, but is able to avoid the problematic reconstruction step in typical VAEs.

Interestingly, we show that a particular PLVM, the RPM, can be effective for SSL (i.e. learning a good high-level representation useful for downstream tasks such as few-shot classification). We would argue that this effectiveness arises precisely because we drop the requirement for easily decoding/sampling "fake-data". This seems to mirror the manner in which state-of-the-art generative models such as diffusions arose from the VAE framework by dropping the requirement for potentially interpretable latent space (Sohl-Dickstein et al., 2015; Ho et al., 2020; Kingma et al., 2021; Song et al., 2021).

Our results have important implications for the interpretation of methods such as InfoNCE. Specifically, InfoNCE was thought to learn good representations by (approximately) maximizing mutual information (Oord et al., 2018). However, recent work has argued that maximizing the true mutual information could lead to arbitrarily entangled representations, as the mutual information is invariant under arbitrary invertible transformations (Tschannen et al., 2019). Instead, they argue that InfoNCE learns good representations because it uses a highly simplified lower bound on the information estimator (Oord et al., 2018) which forms only a loose bound on the true MI. This is highly problematic: Tschannen et al. (2019) argue that better MI estimators give worse representations. Thus, InfoNCE appears to be successful not because it is maximizing MI, but because of ad-hoc choice of simplified mutual information estimator. So what is InfoNCE doing? And how can the success of its simplified mutual information estimator be understood? We show that the InfoNCE objective is equal (up to a constant) to the RPM ELBO. Further, with a deterministic encoder, the RPM ELBO becomes equal to the log marginal likelihood. This would argue that the InfoNCE objective is

better motivated in terms of the RPM ELBO or log marginal likelihood (as they are equal up to a constant in the infinite sample setting), as opposed to the mutual information (as the InfoNCE objective only forms a bound in the same setting).

## 2 Related work

The original version of this paper introduced the key recognition parameterised model idea in 2021, albeit under a different name. However, due to the vagaries of the conference publishing system, follow-up work (Walker et al., 2023b) was published first (at AISTATs). Walker et al. (2023b) introduced the name "Recognition parameterised model", and we agree that this is the right name, so to avoid confusing the literature, we have adopted their terminology. Of course, downstream of the key RPM idea (embodied in the definition of the likelihood in Eq. 10) the papers go in quite different directions, as Walker et al. (2023b) is primarily interested in probabilistic modelling, while we are primarily interested in linking to self-supervised learning. This is particularly evident in three aspects. First, we originally set the approximate posterior to be equal to the recognition distributions, while Walker et al. (2023b) allowed more flexibility in the approximate posteriors, which can slightly improve the quality of their posterior inferences. Second, we made slightly different choices in the "base" distribution (see Appendix A) for details. Third, there are considerable notional differences, in the sense that we defined RPMs in terms of a "recognition joint", while Walker et al. (2023b) defined RPMs in terms of a factor graph. We believe that the recognition joint approach is more intuitive when generalising RPMs to more complex settings, and when using contrastive losses.

A number of papers have taken forward the ideas originally introduced here forward, not just Walker et al. (2023b). First, Walker et al. (2023a) used the RPM in an out-of-distribution setting. Specifically, they are able to retain accurate predictions in an out-of-distribution setting whether *both* the input distribution and the mapping from inputs to labels can change. Second, Möllers et al. (2023) used this approach to obtain uncertainty estimates in a graph SSL setting. Third, Wang et al. (2022) used these ideas to improve sequential recommendation, by improving modelling of user behaviour in the face of sparsity of user-item interactions, uncertainty and a long-tail of items. Fourth, Hasanzadeh et al. (2021) used our approach to introduce principled Bayesian uncertainty estimates in contrastive learning on graph data. In particular, they were able to show improvements in uncertainty estimation, interpretability and predictive performance. Finally, Jelley et al. (2023) used a recognition parameterised model to introduce contrastive methods for metalearning in the few-shot setting (specifically, Eq. 2 in arXiv v0). That said, their work differs in that while they have a recognition parameterised generative model, they do not optimize using VI. Instead, they optimize using the predictive log-likelihood.

Moreover, there are interesting connections to work on connecting self-supervised learning to learning in a linear state-space model (Eysenbach et al., 2024).

Perhaps the closest *prior* (as opposed to follow-up) work is Zimmermann et al. (2021), which also identifies an interpretation of InfoNCE as inference in a principled generative model. Our work differs from (Zimmermann et al., 2021) in that we introduce an RPM and explicitly identify a novel connection between the InfoNCE objective and the RPM ELBO. In addition, their approach requires four restrictive assumptions. First, they assume deterministic encoder. In contrast, all our theory applies to stochastic and deterministic encoders. While we do explicitly consider deterministic encoders in Appendix B, this is only to show that with deterministic encoders, the ELBO bound is tight — all the derivations outside of this very small section (which includes all our key derivations) use fully general encoders. Second, they assume that the encoder is invertible, which is not necessary in our framework. This is particularly problematic as practical encoders commonly used in contrastive SSL are not invertible. Third, they assume that the latent space is unit hypersphere, while in our framework there is no constraint on the latent space. Fourth, they assume the ground truth marginal of the latents of the generative process is uniform, whereas our framework accepts any choice of ground-truth marginal. As such, our framework has considerably more flexibility to include rich priors on complex, structured latent spaces.

Other work looked at the specific case of isolating content from style (von Kügelgen et al., 2021). This work used a similar derivation to that in Zimmermann et al. (2021) with slightly different assumptions. While they still required deterministic, invertible encoders, they relax e.g. uniformity in the latent space.

But because they are working in the specific case of style and content variables, they make a number of additional assumptions on those variables. Importantly, they again do not connect the InfoNCE objective with the ELBO or log marginal likelihood.

Very different methods use noise-contrastive methods to update a VAE prior (Aneja et al., 2020). Importantly, they still use an explicit decoder.

There is a large class of work that seeks to use VAEs to extract useful, disentangled representations (e.g. Burgess et al., 2018; Chen et al., 2018; Kim & Mnih, 2018; Mathieu et al., 2019; Joy et al., 2020). Again, this work differs from our work in that it uses explicit decoders and thus does not identify an explicit link to self-supervised learning.

Likewise, there is work on using GANs to learn interpretable latent spaces (e.g. Chen et al., 2016a). Importantly, GANs learn a decoder (mapping from the random latent space to the data domain). Moreover, GANs use a classifier to estimate a density ratio. However, GANs estimate this density ratio for the data, $x$ and $x'$, whereas InfoNCE, like the methods described here, uses a classifier to estimate a density ratio on the latent space, $z$ and $z'$.

There is work on reinterpreting classifiers as energy-based probabilistic generative models (e.g. Grathwohl et al., 2019), which is related if we view SSL methods as being analogous to a classifier. Our work is very different, if for no other reason than because it is not possible to sample data from a RPM (even using a method like MCMC), because the decoder is written in terms of the unknown true data distribution.

## 3 Background

### 3.1 Variational inference (VI)

In VI, we have observed data, $x$, and latents, $z$, and we specify a prior, $\mathrm{P}(z)$, a likelihood, $\mathrm{P}(x|z)$, and an approximate posterior, $\mathrm{Q}(z|x)$. When the approximate posterior, $\mathrm{Q}(z|x)$ is parameterised by a neural network, the resulting model is known as a variational autoencoder (Kingma & Welling, 2013; Rezende et al., 2014). We then jointly optimize parameters of the prior, likelihood and approximate posterior using the ELBO as the objective,

$$\log \mathrm{P}(x) \geq \mathcal{L}(x) = \mathrm{E}_{\mathrm{Q}(z|x)} \left[ \log \frac{\mathrm{P}(x|z)\,\mathrm{P}(z)}{\mathrm{Q}(z|x)} \right], \tag{1}$$

which bounds the log marginal likelihood, $\log \mathrm{P}(x)$ (as can be shown using Jensen's inequality).

We can rewrite the ELBO as an expected log-likelihood minus a KL-divergence,

$$\mathcal{L}(x) = \mathrm{E}_{\mathrm{Q}(z|x)} \left[ \log \mathrm{P}(x|z) \right] - \mathrm{D}_{\mathrm{KL}} \left( \mathrm{Q}(z|x) \| \mathrm{P}(z) \right). \tag{2}$$

That KL-divergence is very closely related (with a particular choice of prior, $\mathrm{P}(z)$) to the MI (Alemi et al., 2018; Chen et al., 2016b), so the KL-divergence can be understood intuitively as reducing the MI between data, $x$ and latent, $z$. However this connection between the ELBO and MI is not relevant for our work. In particular, the ELBO can be interpreted as minimizing the MI between data and latents, whereas InfoNCE maximizes the MI between different latent variables in a structured model.

### 3.2 InfoNCE

In InfoNCE (Oord et al., 2018), there are two data items, $x$ and $x'$. Oord et al. (2018) initially describes a time-series setting where for instance $x$ is the previous datapoint and $x'$ is the current datapoint. But one can also consider other contexts where $x$ and $x'$ are different augmentations or patches of the same underlying image (Chen et al., 2020). In all cases, there are strong dependencies between $x$ and $x'$, and the goal is to capture these dependencies with latent representations, $z$ and $z'$, formed by passing $x$ and $x'$ through neural network encoders. All our derivations consider fully general encoders, $\mathrm{R}_\phi(z|x)$ and $\mathrm{R}_\phi(z'|x')$ which could be stochastic or deterministic (see Sec. B, where we discuss this distinction in-depth). Note that we write

these encoders with explicit $\phi$ subscripts to emphasise that these are user-specified conditional distributions, often a Gaussians over $z$ or $z'$ with means and variances specified by applying a neural network to $x$ or $x'$.

Importantly, we do not just use R to denote the encoder distributions, $\mathrm{R}_\phi\left(z|x\right)$ and $\mathrm{R}_\phi\left(z'|x'\right)$. We generalise R to denote the joint distribution arising from taking the true / empirical training data distribution, $\mathrm{R}_{\mathrm{base}}\left(x, x'\right)$ (see Appendix A) and encoding using $\mathrm{R}_\phi\left(z|x\right)$ and $\mathrm{R}_\phi\left(z'|x'\right)$. Thus, the joint distribution of all random variables under R can be written,

$$\mathrm{R}\left(x, x', z', z\right) = \mathrm{R}_\phi\left(z|x\right)\mathrm{R}_\phi\left(z'|x'\right)\mathrm{R}_{\mathrm{base}}\left(x, x'\right). \tag{3}$$

We can obtain marginals and conditionals of this joint, such as $\mathrm{R}\left(z', z\right)$, $\mathrm{R}\left(z'\right)$, $\mathrm{R}\left(z\right)$, $\mathrm{R}\left(z'|z\right)$ in the usual manner — by marginalising and/or applying Bayes theorem. For instance,

$$\mathrm{R}\left(z, z'\right) = \int dx\ dx'\ \mathrm{R}_\phi\left(z|x\right)\mathrm{R}_\phi\left(z'|x'\right)\mathrm{R}_{\mathrm{base}}\left(x, x'\right). \tag{4}$$

Note that $\mathrm{R}\left(z, z'\right)$ also implicitly depends on $\phi$. We use the subscript $\phi$ in $\mathrm{R}_\phi\left(z|x\right)$ to remind the reader that $\mathrm{R}_\phi\left(z|x\right)$ is directly parameterised by e.g. a neural network with weights $\phi$, and we omit the subscript in e.g. $\mathrm{R}\left(z, z'\right)$ to indicate that its dependence on $\phi$ is only implicit (through Eq. 4).

The InfoNCE objective was originally motivated as maximizing the mutual information between latent representations,

$$\mathrm{MI} = \mathrm{E}_{\mathrm{R}(z,z')}\left[\log\frac{\mathrm{R}\left(z'|z\right)}{\mathrm{R}\left(z'\right)}\right] = \mathrm{E}_{\mathrm{R}(z,z')}\left[\log\frac{\mathrm{R}\left(z', z\right)}{\mathrm{R}\left(z'\right)\mathrm{R}\left(z\right)}\right]. \tag{5}$$

Of course, observed $x$ and $x'$ exhibit dependencies as they are e.g. the previous and current datapoints in a timeseries, or different augmentations of the same underlying image. Thus, $z$ and $z'$ must also exhibit dependencies under $\mathrm{R}\left(z, z'\right)$. As the mutual information is difficult to compute directly, InfoNCE uses a bound, $\mathcal{I}_N(\theta, \phi)$, based on a classifier that uses $f_\theta$ with parameters $\theta$ to distinguish positive samples (i.e. the $z'$ paired with the corresponding $z$) from negative samples (i.e. $z'_j$ drawn from the marginal distribution and unrelated to $z$ or to the underlying data; see Poole et al., 2019 for further details),

$$\mathrm{MI} \geq \mathcal{I}_N = \mathrm{E}\left[\log\frac{f_\theta(z, z')}{f_\theta(z, z') + \sum_{j=1}^N f_\theta(z, z'_j)}\right] + \log N. \tag{6}$$

Here, the expectation is taken over $\mathrm{R}\left(z, z'\right)\prod_j \mathrm{R}\left(z'_j\right)$, and we use this objective to optimize $\theta$ (the parameters of $f_\theta$) and $\phi$ (the parameters of the encoder). There are two source of slack in this bound, arising from finite $N$ and a restrictive choice of $f$. To start, we can reduce but not in general eliminate slack by taking the limit as $N$ goes to infinity, (Oord et al., 2018),

$$\mathrm{MI} > \mathcal{I}_\infty(\theta, \phi) \geq \mathcal{I}_N(\theta, \phi). \tag{7}$$

The bound only becomes tight if we additionally optimize an arbitrarily flexible $f$ (Oord et al., 2018). If as usual, we have a restrictive parametric family for $f$, then the bound does not in general become tight (Oord et al., 2018). In reality InfoNCE does indeed use a highly restrictive class of function for $f$, which can be expected to give a loose bound on the MI (Oord et al., 2018),

$$f_\theta(z, z') = \exp\left(z^T \theta z'\right), \tag{8}$$

where $\theta$ for this particular function is a matrix. Note additionally that for this particular functional form for $f_\theta$ to work well, we usually need to restrict $z$ and $z'$ to have unit-norm.

This raises a critical question: if our goal is really to maximize the bound on the MI, why not use a more flexible $f_\theta$? The answer that our goal is not ultimately to maximize the MI. Our goal is ultimately to learn a good representation, and MI is merely a means to that end. Further, Tschannen et al. (2019) argue that optimizing the true MI is likely to lead give poor repesentations, as the MI is invariant to arbitrary invertible transformations that can entangle the representation. They go on to argue that it is

precisely the restrictive family of functions, corresponding to a loose bound on the MI, that encourages good representations. Tschannen et al. (2019) thus raise an important question: does it really make sense to motivate an objective that works (the InfoNCE objective) as a loose bound on an objective that does not work (the mutual information)? We offer an alternative motivation by showing that the InfoNCE objective is equal (up to a constant) to the log marginal likelihood under a particular choice of prior and with a deterministic encoder.

Note that this loss has been used in downstream settings, such as SimCLR (Chen et al., 2020).

## 4 Recognition Parameterised Models

An RPM is a specific type of probabilistic latent variable model (PLVM) where the likelihood is written in terms of a recognition model. All RPMs have a structured graphical model with multiple observations. We start with perhaps the simplest RPM, where observations are a pair, $(x, x')$, e.g. two augmentations of the same image, or two different but adjacent frames in a video. In this simple example, we consider a model with latent variable, $z$ associated with $x$ and $z'$ associated with $x'$. The generative probability in our probabilistic generative model can be factorised as,

$$\mathrm{P}\left(x, x', z', z\right) \equiv \mathrm{P}\left(x|z\right)\mathrm{P}\left(x'|z'\right)\mathrm{P}_\theta\left(z, z'\right). \tag{9}$$

Here, we write the prior as $\mathrm{P}_\theta\left(z, z'\right)$ to emphasise that this could be a user-specified distribution with learned parameters, $\theta$. Importantly, P implies a valid joint distribution (Eq. 9) over all random variables $(x, x', z', z)$, so any time we write a distribution with P, we mean a marginal / conditional of that model joint (Eq. 9).

Importantly, we have yet to define the likelihood, and it is the likelihood that makes it an RPM. In particular, in an RPM, the likelihood is defined in terms of Bayesian inference in the recognition joint, R, formed by taking the true/empirical data distribution, and encoding using $\mathrm{R}_\phi\left(z|x\right)$ and $\mathrm{R}_\phi\left(z'|x'\right)$ (Eq. 3). We can use Bayes in the recognition joint to obtain $\mathrm{R}\left(x|z\right)$ and $\mathrm{R}\left(x'|z'\right)$. We then decide to use $\mathrm{R}\left(x|z\right)$ and $\mathrm{R}\left(x'|z'\right)$ as the likelihoods in our generative model,

$$\mathrm{P}\left(x|z\right) \equiv \mathrm{R}\left(x|z\right) = \frac{\mathrm{R}_\phi\left(z|x\right)\mathrm{R}_{\mathrm{base}}\left(x\right)}{\mathrm{R}\left(z\right)}, \tag{10a}$$

$$\mathrm{P}\left(x'|z'\right) \equiv \mathrm{R}\left(x'|z'\right) = \frac{\mathrm{R}_\phi\left(z'|x'\right)\mathrm{R}_{\mathrm{base}}\left(x'\right)}{\mathrm{R}\left(z'\right)}. \tag{10b}$$

Here, we have written $\mathrm{P}\left(x|z\right) \equiv \mathrm{R}\left(x|z\right)$ to denote that we are choosing the generative likelihood, $\mathrm{P}\left(x|z\right)$, to be equal to $\mathrm{R}\left(x|z\right)$ from the recognition joint. We have written Bayes theorem with an $=$ because that is not a choice. All distributions written with a R represent a coherent joint distribution (Eq. 3) and hence $\mathrm{R}\left(x|z\right)$ must be given by Bayes theorem applied to that joint distribution.

The normalizing constants, $\mathrm{R}\left(z\right)$ and $\mathrm{R}\left(z'\right)$, are,

$$\mathrm{R}\left(z\right) = \int dx\,\mathrm{R}_\phi\left(z|x\right)\mathrm{R}_{\mathrm{base}}\left(x\right), \tag{11a}$$

$$\mathrm{R}\left(z'\right) = \int dx'\,\mathrm{R}_\phi\left(z'|x'\right)\mathrm{R}_{\mathrm{base}}\left(x'\right). \tag{11b}$$

See Appendix A for further details.

### 4.1 VI in RPMs

Now, we substitute these generative probabilities into the VI objective (i.e. the ELBO), using an approximate posterior, $\mathrm{Q}_\psi\left(z, z'|x, x'\right)$,

$$\mathcal{L}(x, x') = \mathrm{const} + \mathrm{E}_\mathrm{Q}\left[\log \frac{\mathrm{P}_\theta\left(z, z'\right)}{\mathrm{R}\left(z\right)\mathrm{R}\left(z'\right)} + \log \frac{\mathrm{R}_\phi\left(z|x\right)\mathrm{R}_\phi\left(z'|x'\right)}{\mathrm{Q}_\psi\left(z, z'|x, x'\right)}\right]. \tag{12}$$

Here,

$$\text{const} = \log \left( \text{R}_{\text{base}} \left( x \right) \text{R}_{\text{base}} \left( x' \right) \right), \tag{13}$$

is constant because $\text{R}_{\text{base}} \left( x \right)$ and $\text{R}_{\text{base}} \left( x' \right)$ are either the true data marginals or the empirical marginals (Appendix A). In either case, these distributions do not depend on the prior parameters, $\theta$, the recognition parameters, $\phi$, or the approximate posterior parameters, $\psi$.

### 4.2  InfoNCE as a RPM

Now, we choose the approximate posterior to be equal to the corresponding conditional of the recognition joint,

$$\text{Q} \left( z, z' | x, x' \right) \equiv \text{R} \left( z, z' | x, x' \right) = \text{R}_{\phi} \left( z | x \right) \text{R}_{\phi} \left( z' | x' \right) \tag{14}$$

Note that Walker et al. (2023b) do not make this choice. Instead, they allow the approximate posterior to be optimized separately; on the probabilistic modelling tasks they consider, their choice is likely to lead to slightly improved performance. Substituting this choice for the approximate posterior, the ELBO simplifies considerably,

$$\mathcal{L}(x, x') = \text{const} + \text{E}_{\text{R}_{\phi}(z|x) \, \text{R}_{\phi}(z'|x')} \left[ \log \frac{\text{P}_{\theta} \left( z, z' \right)}{\text{R} \left( z \right) \text{R} \left( z' \right)} \right]. \tag{15}$$

Additionally, we consider the expected loss, taking the expectation over the true/empirical data distribution, $\text{R}_{\text{base}} \left( x, x' \right)$,

$$\mathcal{L} = \text{E}_{\text{R}_{\text{base}}(x,x')} \left[ \mathcal{L}(x, x') \right] = \text{const} + \text{E}_{\text{R}(z,z')} \left[ \log \frac{\text{P}_{\theta} \left( z, z' \right)}{\text{R} \left( z \right) \text{R} \left( z' \right)} \right]. \tag{16}$$

where $\text{R} \left( z, z' \right)$ is the relevant marginal of the recognition joint (Eq. 4).

### 4.3  The RPM ELBO can be written as the mutual information minus a KL divergence

To get an intuitive understanding of the ELBO we take Eq. (16) and add and subtract $\text{E}_{\text{R}(z,z')} \left[ \log \text{R} \left( z, z' \right) \right]$,

$$\mathcal{L} = \text{const} + \text{E}_{\text{R}(z,z')} \left[ \log \frac{\text{R} \left( z, z' \right)}{\text{R} \left( z \right) \text{R} \left( z' \right)} \right] + \text{E}_{\text{R}(z,z')} \left[ \log \frac{\text{P}_{\theta} \left( z, z' \right)}{\text{R} \left( z, z' \right)} \right]. \tag{17}$$

The first term is the mutual information between $z$ and $z'$ under the recognition joint, R (Eq. 5), and the second term is a KL-divergence,

$$\mathcal{L} = \text{const} + \text{MI} - \text{D}_{\text{KL}} \left( \text{R} \left( z, z' \right) \| \text{P}_{\theta} \left( z, z' \right) \right). \tag{18}$$

This objective therefore encourages large mutual information between $z$ and $z'$ under R (Eq. 4), while encouraging $\text{R} \left( z, z' \right)$ to lie close to the prior, $\text{P}_{\theta} \left( z, z' \right)$.

### 4.4  Under the optimal prior, the RPM ELBO is equal to the mutual-information (up to a constant)

Looking at Eq. (18), the only term that depends on the prior, $\text{P}_{\theta} \left( z', z \right)$, is the negative KL-divergence. As such, maximizing $\mathcal{L}$ with respect to the parameters of $\text{P}_{\theta} \left( z', z \right)$ is equivalent to minimizing $\text{D}_{\text{KL}} \left( \text{R} \left( z, z' \right) \| \text{P}_{\theta} \left( z, z' \right) \right)$. Of course, the minimal KL-divergence of zero is obtained when,

$$\text{P}^* \left( z, z' \right) = \text{R} \left( z, z' \right). \tag{19}$$

For this optimal prior, the KL-divergence is zero, so the ELBO reduces to just the mutual information between $z$ and $z'$ (and a constant),

$$\mathcal{L}_{\text{MI}} = \text{const} + \text{MI}. \tag{20}$$

### 4.5 Under a particular prior, the RPM ELBO is equal to the infinite-sample InfoNCE objective (up to a constant)

Recent work has argued that the good representation arising from InfoNCE cannot be from maximizing mutual information alone, because the mutual information is invariant under arbitrary invertible transformations (Tschannen et al., 2019; Li et al., 2021). Instead, the good properties must arise somehow out of the fact that the InfoNCE objective forms only a loose bound on the true MI, even in the infinite sample limit Eq. (7). In contrast, here we show that the infinite-sample InfoNCE objective is equal (up to a constant) to the ELBO (or log marginal likelihood for deterministic encoders) for a specific choice of prior. In particular, we choose the prior on $z$ implicitly (through $\mathrm{R}(z)$), and we choose the distribution over $z'$ conditioned on $z$ to be given by an energy based model with an unrestricted coupling function, $f_\theta(z, z')$ (we could of course use Eq. 8 for $f_\theta$),

$$\mathrm{P}^{\mathrm{InfoNCE}}(z) = \mathrm{R}(z) \tag{21a}$$

$$\mathrm{P}_\theta^{\mathrm{InfoNCE}}(z'|z) = \frac{1}{Z(z)} \mathrm{R}(z') f_\theta(z, z'). \tag{21b}$$

The normalizing constant, $Z(z)$, is

$$Z(z) = \int dz' \, \mathrm{R}(z') f_\theta(z, z') = \mathrm{E}_{\mathrm{R}(z')}[f_\theta(z, z')]. \tag{22}$$

Substituting these choices into Eq. (16), and cancelling $\mathrm{R}(z)\,\mathrm{R}(z')$, the average ELBO or log marginal likelihood becomes,

$$\mathcal{L}_{\mathrm{InfoNCE}} = \mathrm{const} + \mathrm{E}_{\mathrm{R}(z,z')}\left[\log f_\theta(z, z') - \log Z(z)\right], \tag{23}$$

and substituting for $Z(z)$ gives,

$$\mathcal{L}_{\mathrm{InfoNCE}} = \mathrm{const} + \mathrm{E}_{\mathrm{R}(z,z')}\left[\log f_\theta(z, z')\right] - \mathrm{E}_{\mathrm{R}(z)}\left[\log \mathrm{E}_{\mathrm{R}(z')}[f_\theta(z, z')]\right]. \tag{24}$$

Following Wang & Isola (2020) and Li et al. (2021) the right hand side can be identified as the infinite sample InfoNCE objective that we introduced in Sec. 3.2,

$$\mathrm{MI} > \mathcal{I}_\infty = \mathrm{const} + \mathcal{L}_{\mathrm{InfoNCE}}. \tag{25}$$

Therefore, in this choice of model, the ELBO, $\mathcal{L}_{\mathrm{InfoNCE}}$, is equal to the infinite-sample InfoNCE objective, $\mathcal{I}_\infty$, up to a constant. This would argue that the InfoNCE objective has a closer link to the ELBO, $\mathcal{L}_{\mathrm{InfoNCE}}$, than it does to the MI, as the infinite-sample InfoNCE objective, $\mathcal{I}_\infty(\theta, \phi)$ is equal (up to a constant) to log marginal likelihood (with a determinstic encoder; Appendix C) or the ELBO with a stochastic encoder. Whereas the infinite-sample InfoNCE objective only gives a bound on the MI.

Of course, this is all in the infinite sample setting: we have an infinite-sample expression for the InfoNCE objective (which forms a bound on the MI) and we have an infinite-sample expression for the ELBO/log marginal likelihood. However, the infinite-sample InfoNCE objective and the infinite-sample ELBO are equal up to additive constants, so these are really just different interpretations of the same quantity. The question then becomes how to develop a finite-sample bound on this single quantity with two slightly different interpretations. As we really only have a single quantity, we only need a single finite-sample bound. That single finite-sample bound would of course apply equally to both the InfoNCE and ELBO interpretations. Ultimately, we choose to use the finite-sample bound originally described in the InfoNCE framework (Oord et al., 2018), but as discussed above, that bound would of course apply equally to both the InfoNCE and ELBO interpretations.

## 5 Experimental results

Our primary results are theoretical: in connecting the ELBO/log marginal likelihood and mutual information, and in showing that the InfoNCE objective with a restricted choice of $f_\theta$ makes more sense as a bound on

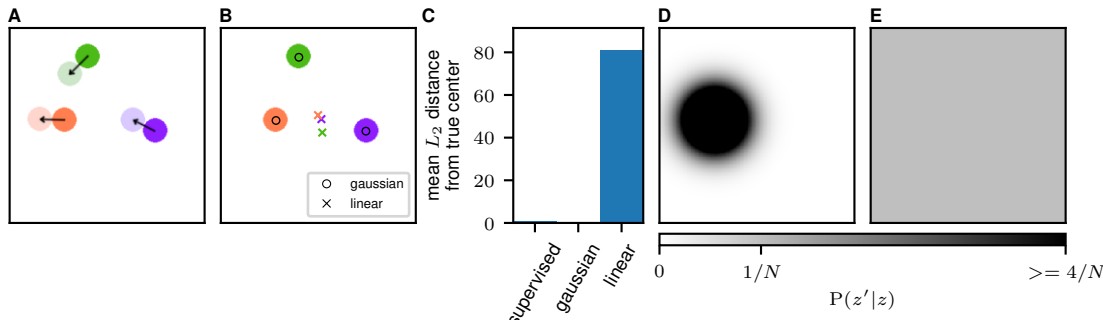

Figure 1: Results of the moving balls experiment. **A**) Example of the motion between consecutive frames. The balls move by a full diameter in a semi-random direction. **B**) Locations of the extracted ball centres, after supervised linear decoding. The standard InfoNCE setup fails to extract correct locations. **C**) The mean distance from the extracted and true centres of the balls for a supervised method, InfoNCE with a Gaussian discriminator after supervised decoding and InfoNCE with a linear discriminator after supervised decoding. **D**) Probability distribution for the next location of the coral ball in **A** according to an encoder trained with a Gaussian discriminator. **E**) Probability distribution for the next location of the same ball according to an encoder trained with a linear discriminator.

the log marginal likelihood than on the MI. At the same time, our approach encourages a different way of thinking about how to set up contrastive SSL methods, in terms of Bayesian priors. As an example, we considered a task in which the goal was to extract the locations of three moving balls, based on videos of these balls bouncing around in a square (Fig. 1A; Appendix D).

Critically, we want the latent space, $z$ and $z'$, to mirror the underlying true latent space as closely as possible. We take the latent spaces to be 6 dimensional, representing the x and y positions of the 3 balls. Critically, it does not make sense to impose a unit norm constraint on this latent space, as such a constraint does not exist in the real latent space. This contrasts with the usual InfoNCE setup, where $z$ and $z'$ are usually taken to be unit norm.

To resolve this difficulty, we consider the InfoNCE-like setup described in Sec. 4.5. Our prior is given by Eq. (21), with $\mathrm{R}\,(z)$ and $\mathrm{R}\,(z')$ defined by Eq. (11). The freedom in this setup is given by the choice of $f_\theta$. Naively applying the usual InfoNCE choice of $f_\theta$ without unit-norm constraints (using Eq. 8), failed (linear in Fig. 1BC), because we did not correctly encode prior information about the structure of the problem. Critically, our prior is that for the adjacent frames, the locations extracted by the network will be close, while for random frames, the locations extracted by the network will be far apart. The linear estimator in Eq. (8) is not suitable for extracting the proximity of the ball locations, so it fails (linear in Fig. 1 BC). In particular, it corresponds to a non-sensical prior over $z'$ given $z$,

$$\mathrm{P}^{\mathrm{InfoNCE}}\,(z'|z) = \frac{1}{Z(z)}\,\mathrm{R}\,(z')\,f_\theta(z, z') \propto \exp\left(z^T \theta z'\right) \tag{26}$$

(where we have taken $\mathrm{Q}_\phi\,(z')$ defined by Eq. 11 to be approximately uniform purely for the purposes of building intuition). This prior will encourage $z'$ to be very large (specifically, $z'$ should have a large dot-product with $z^T\theta$). Instead, we would like a prior that encodes our knowledge that $z'$ is likely to be close to $z$. We can get such a prior by using a Gaussian RBF form for $f_\theta$,

$$f_\theta(z, z') = \exp\left(-\tfrac{1}{2\theta^2}(z - z')^2\right). \tag{27}$$

where the only parameter, $\theta$, is a scalar learned lengthscale. Critically, this choice of $f_\theta$ is natural and obvious if we take a probabilistic generative view of the problem (with a uniform $\mathrm{Q}_\phi\,(z')$, this corresponds to a Gaussian conditional, $\mathrm{P}^{\mathrm{InfoNCE}}_{\theta,\phi}\,(z'|z)$). Because this is the natural and obvious choice of prior, it works well even without unit-norm constraints (which as discussed, do not make sense in this setting).

## 6    Conclusions

In conclusion, we have developed a new family of probabilistic generative model, the "recognition parameterised model". With a determinstic recognition model, the RPM ELBO is equal to the marginal likelihood (Appendix C). For the optimal prior, the RPM ELBO is equal (up to a constant) to the mutual information, and with a particular choice prior, the RPM ELBO is equal (up to a constant) to the infinite-sample InfoNCE objective (up to constants). In contrast, the infinite-sample InfoNCE forms only a loose bound on the true MI, which would argue that the InfoNCE objective might be better motivated as the RPM ELBO. As such, we unify contrastive semi-supervised learning with generative self-supervised learning (or unsupervised learning). Finally, we provide a principled framework for using simple parametric models in the latent space to enforce disentangled representations, and our framework allows us to use Bayesian intuition to form richer priors on the latent space.

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

# A    Choices for the base distribution

There are three choices of base marginals, $R_{\text{base}}(x)$ and $R_{\text{base}}(x')$ in an RPM, corresponding to three different variants of the RPM:

1. The empirical marginal RPM.

2. The true marginal RPM.

3. The estimated marginal RPM.

It will turn out that at least the choice between 1 and 2 makes little difference, as they lead to exactly the same expression for the ELBO (compare Eq. 29 and Eq. 37), albeit with a slightly different interpretation.

## A.1    Empirical marginal RPM

The empirical marginal RPM was introduced in Walker et al. (2023b), and uses,

$$R_{\text{base emp}}(x) = \tfrac{1}{N} \sum_i \delta(x - x_i) \tag{28a}$$

$$R_{\text{base emp}}(x') = \tfrac{1}{N} \sum_i \delta(x' - x'_i). \tag{28b}$$

where $i$ indexes training data points. This has the advantage that $R_{\text{base emp}}(x)$ and $R_{\text{base emp}}(x')$ are known. Thus, $R(z)$ and $R(z')$ can be evaluated exactly as a mixture distribution,

$$R(z) = \tfrac{1}{N} \sum_i R(z|x_i) \tag{29a}$$

$$R(z') = \tfrac{1}{N} \sum_i R(z'|x'_i) \tag{29b}$$

In this setting, we know $R_{\text{base emp}}(x)$, $R(z)$ and (of course) $R_\phi(z|x)$, so we can exactly evaluate the probability density for the recognition parameterised likelihood (Eq. 10). As such, we can sample from the model distribution over data, $P(x, x')$.

However, the empirical marginal viewpoints does has important issues. Specifically,

$$P(x, x') = \int dz dz' \, P(z, z') \, P(x|z) \, P(x'|z') \tag{30}$$

Substituting Eq. (10),

$$P(x, x') = R_{\text{base emp}}(x) \, R_{\text{base emp}}(x') \, F(x, x') \tag{31}$$

where,

$$F(x, x') = \int dz dz' \, P(z, z') \frac{R_\phi(z|x)}{R(z)} \frac{R_\phi(z'|x')}{R(z')}. \tag{32}$$

Substituting the empirical marginal form for $R_{\text{base emp}}(x)$ and $R_{\text{base emp}}(x)$ (Eq. (28)),

$$P(x, x') = \left( \tfrac{1}{N} \sum_{i=1}^N \delta(x - x_i) \right) \left( \tfrac{1}{N} \sum_{j=1}^N \delta(x' - x'_j) \right) F(x_i, x'_j) \tag{33}$$

This distribution only places probability density/mass at values of $x$ and $x'$ that were actually observed in the data, i.e. $x \in \{x_i\}_{i=1}^N$ and $x' \in \{x'_i\}_{i=1}^N$. In most cases we do not expect the exact test value of $x$ and $x'$ to be in the training set, and this model assigns those test points zero probability density. Indeed, for

absolutely continuous distributions, we *never* expect the exact test value of $x$ and $x'$ to be in the training set.

Additionally, we can again connect this form to InfoNCE-like objectives. In particular, as the probability density over $(x, x')$ above places mass only at a finite number of training points, we can equivalently write it as a probability mass function over indicies $(i, j)$ describing which datapoint was chosen. The resulting probability mass function is,

$$\text{P}(i, j) = P_{ij} = \tfrac{1}{N^2} F(x_i, x'_j). \tag{34}$$

Critically, the implied conditional, $\text{P}(j|i)$, in essence embodies a classification problem: classifying which of the $N$ training points, $x'_j$, is associated with a particular $x_i$. And that classification problem is almost exactly the InfoNCE classification problem.

Finally, this implies that even if we do sample from $\text{P}(x, x')$, we are definitely going to end up with $x$ and $x'$ from the training dataset. As such, "sampling" is just "matching up" potentially consistent $x$ and $x'$ in the training set, and is thus quite different from sampling in a proper generative model.

## A.2   True marginal RPM

While the empirical marginal RPM has many advantages, it does assign zero probability to any test points that did not turn up in the training data. To resolve this issue, we could instead use the true data distribution for $\text{R}_{\text{base}}$. It seems that this would be problematic as while the true data distribution exists, it is almost always unknown (and even unknowable). However, remember that to perform inference and learn the parameters, we never need to evaluate probability density of the true data distribution, as the $\text{R}_{\text{base}}(x')$ and $\text{R}_{\text{base}}(x)$ terms cancel out of the ELBO (Eq. 12). Instead, we just need to be able to sample $\text{R}(z, z')$, which can easily be achieved by taking $(x', x)$ from the real data, and encoding using $\text{R}_\phi(z|x)$ and $\text{R}_\phi(z'|x')$. Of course, not knowing the true data distribution does prevent us from sampling $(x', x)$ from the model in a true marginal RPM. But as discussed in the Introduction, we are not interested in sampling data from the model: if we were, we would use e.g. a diffusion model. Instead, we are interested in extracting useful structured, high-level representations of data, and for that purpose, all we need to be able to do is to encode using $\text{R}_\phi(z|x)$ and $\text{R}_\phi(z'|x')$ and optimize the parameters of these encoders.

In practice, the empirical and true marginal settings are almost identical. The only minor difference is that in the empirical marginal setting, $\text{R}(z)$ and $\text{R}(z')$ can be evaluated exactly (Eq. 29), whereas in the true marginal setting, $\text{R}(z)$ and $\text{R}(z')$ can only be estimated. Critically, though the numerical value of the exact $\text{R}(z)$ and $\text{R}(z')$ in the empirical marginal setting (Eq. 29) is exactly the same as the true marginal estimate of these quantities. Speicifically, in the true marginal setting,

$$\text{R}(z) = \int dx \, \text{P}_{\text{true}}(x) \, \text{R}_\phi(z|x) \tag{35a}$$

$$\text{R}(z') = \int dx' \, \text{P}_{\text{true}}(x') \, \text{R}_\phi(z'|x'). \tag{35b}$$

However, we can rewrite these integrals as expectations,

$$\text{R}(z) = \text{E}_{\text{P}_{\text{true}}(x)} \left[ \text{R}(z|x) \right] \tag{36a}$$

$$\text{R}(z') = \text{E}_{\text{P}_{\text{true}}(x')} \left[ \text{R}(z'|x') \right] \tag{36b}$$

We can estimate these expectations using samples from $\text{P}_{\text{true}}(x)$ and $\text{P}_{\text{true}}(x')$. Of course, the data itself gives samples from $\text{P}_{\text{true}}(x)$ and $\text{P}_{\text{true}}(x')$. Thus, even in the true-marginal setting, we would estimate $\text{R}(z)$ and $\text{R}(z')$ using the same sum over datapoints used in the empirical marginal setting (i.e. Eq. 29),

$$\text{R}(z) \approx \tfrac{1}{N} \sum_i \text{R}(z|x_i) \tag{37a}$$

$$\text{R}(z') \approx \tfrac{1}{N} \sum_i \text{R}(z'|x'_i) \tag{37b}$$

The only difference is in the interpretation of this sum. Here (in the true marginal setting), the sum *estimates* the true value of $\text{R}(z)$ and $\text{R}(z')$, while in the empirical marginal setting (Eq. 29), the sum *is* the true value.

### A.3 Estimated marginal RPM

We could of course learn $R_{\text{base est}}(x)$ and $R_{\text{base est}}(x')$. That results in a model where we can both evaluate the probability density, and sample $P(x', x)$. However, this involves training a generative model for complex data, such as images. While that might seem to obviate the whole point of the exercise, there is the potential for estimated marginal RPMs to be useful. In particular, the estimated marginal RPM only requires training a generative model for $x$ and $x'$ separately (note that the ELBO in Eq. 12 depends only on the marginals, $R(x)$ and $R(x')$, and not on $R(x, x')$. Thus, it may be possible to use the estimated marginal to "stitch together" samples of the joint, $(x, x')$ from a generative model which can sample only from marginals.

## B   Technical details when using deterministic recognition models

While we do not have to, we could follow the standard self-supervised setup, which in effect uses Dirac-delta recognition models,

$$R_\phi\left(z\,|x\,\right) = \delta\left(z\,-g_\phi(x)\right), \tag{38}$$

$$R_\phi\left(z'|x'\right) = \delta\left(z'-g'_\phi(x')\right). \tag{39}$$

That raises the question of when our objectives make sense. To answer these questions, we need to carefully understand the measure-theory underlying the expressions in the main text. In particular, our objectives are all ultimately written in terms of KL-divergences for distributions over the latent variables, $(z, z')$. We take $z \in \mathcal{Z}$ and $z' \in \mathcal{Z}'$, and consider probability measures $\mu$ and $\nu$, where the set underlying the measure space is $\mathcal{Z} \times \mathcal{Z}'$. Thus, the KL-divergences in the objective can be written in measure-theoretic notation as,

$$D_{\text{KL}}\left(\nu\|\mu\right) = \int d\nu \log \frac{d\nu}{d\mu}, \tag{40}$$

(e.g. see Wild et al. 2022). The critical term is $d\nu/d\mu$, which is known as the Radon-Nikodym derivative. This derivative is defined when we can write $\nu$ in terms of a function $f : \mathcal{Z} \times \mathcal{Z}' \to [0, \infty)$.

$$\frac{d\nu}{d\mu} = f \tag{41}$$

$$\nu(A) = \int_A f d\mu \tag{42}$$

where $A \subseteq \mathcal{Z} \times \mathcal{Z}'$. In that case, $f$ is uniquely defined up to a $\mu$-null set (informally, if $\nu(A) = \mu(A) = 0$ then $f$ is not uniquely defined for $(z, z') \in A$). Now, we can consider the two KL-divergence terms in Eq. (17).

The first KL-divergence term is,

$$E_{R(z,z')}\left[\log \frac{R\left(z, z'\right)}{R\left(z\right)R\left(z'\right)}\right] = D_{\text{KL}}\left(R\left(z, z'\right)\|R\left(z\right)R\left(z'\right)\right). \tag{43}$$

In this case, we can write,

$$R\left(z, z'\right) = f(z, z')R\left(z\right)R\left(z'\right), \tag{44}$$

so the Radon-Nikodym derivative exists, irrespective of the form of $R_\phi\left(z|x\right)$ and $R_\phi\left(z'|x'\right)$. This also implies that in Section 4.4, where we use the optimal prior, the objective (Eq. 20) is always well-defined.

The second KL-divergence term is,

$$E_{R(z,z')}\left[\log \frac{P_\theta\left(z, z'\right)}{R\left(z, z'\right)}\right] = -D_{\text{KL}}\left(R\left(z, z'\right)\|P\left(z, z'\right)\right). \tag{45}$$

If we use an optimal prior, this KL-divergence cancels. However, if we do not use the optimal prior, but instead want to use a parametric prior, this term can cause problems. The issue is that if we choose a

standard form for the prior, such as a multivariate Gaussian, then $\mathrm{P}\left(z, z'\right)$ will be absolutely continuous. However if we have deterministic $\mathrm{R}_\phi\left(z|x\right)$ and $\mathrm{R}_\phi\left(z'|x'\right)$, then $\mathrm{R}\left(z, z'\right)$ can be supported on a measure-zero subspace, and hence it will not be absolutely continuous. This for instance might happen if $(x, x')$ and $(z, z')$ are real vector spaces, with $x$ lower dimensional than $z$ and/or $x'$ lower dimensional than $z'$. One way to resolve this issue would be to set,

$$\mathrm{R}_\phi\left(z|x\right) = \mathcal{N}\left(g_\phi(x), \epsilon\mathbf{I}\right) \tag{46}$$

$$\mathrm{R}_\phi\left(z'|x'\right) = \mathcal{N}\left(g'_\phi(x'), \epsilon\mathbf{I}\right) \tag{47}$$

where $0 < \epsilon$ is arbitrarily small. In that case, $\mathrm{R}_\phi\left(z|x\right)\mathrm{R}_\phi\left(z'|x'\right)$ and hence $\mathrm{R}\left(z, z'\right)$ are absolutely continuous and non-zero everywhere so the necessary Radon-Nikodym derivative exists.

Also see Nielsen et al. (2020) which also uses deterministic approximate posteriors in the VAE setting, and compares them against normalizing flows. Note however that they work with quantities such as $\mathrm{E}_{\mathrm{Q}(z|x)}\left[\log \mathrm{P}\left(x|z\right)/\mathrm{Q}\left(z|x\right)\right]$ which lack a rigorous measure-theoretic formulation, as the distributions in the ratio are over different variables: $x$ in the numerator and $z$ in the numerator. As such, approach presented here which does allow a measure-theoretic formulation is likely to be preferable.

## C  When we use deterministic recognition models, the ELBO is equal to the (marginal) log likelihood)

The marginal likelihood, $\mathrm{P}\left(x, x'\right)$, is,

$$\mathrm{P}\left(x, x'\right) = \int dz dz'\, \mathrm{P}\left(x|z\right) \mathrm{P}\left(x'|z'\right) \mathrm{P}\left(z, z'\right). \tag{48}$$

Substituting the recognition parameterised likelihoods,

$$\log \mathrm{P}\left(x, x'\right) = \log \int dz dz'\, \mathrm{P}\left(z, z'\right) \frac{\mathrm{R}_\phi\left(z|x\right)\mathrm{R}_{\mathrm{base}}\left(x\right)}{\mathrm{R}\left(z\right)} \frac{\mathrm{R}_\phi\left(z'|x'\right)\mathrm{R}_{\mathrm{base}}\left(x'\right)}{\mathrm{R}\left(z'\right)} \tag{49}$$

$$= \mathrm{const} + \log \mathrm{E}_{\mathrm{R}_\phi\left(z|x\right)\mathrm{R}_\phi\left(z'|x'\right)}\left[\frac{\mathrm{P}\left(z, z'\right)}{\mathrm{R}\left(z\right)\mathrm{R}\left(z'\right)}\right] \tag{50}$$

where, as usual $\mathrm{const} = \log\left(\mathrm{R}_{\mathrm{base}}\left(x\right)\mathrm{R}_{\mathrm{base}}\left(x'\right)\right)$. The recognition models in the expectation are deterministic (Eq. 38), $z = g_\phi(x)$ and $z' = g'_\phi(x')$, so we can easily evaluate the expectation,

$$\log \mathrm{P}\left(x, x'\right) = \mathrm{const} + \log \frac{\mathrm{P}\left(z, z'\right)}{\mathrm{R}\left(z\right)\mathrm{R}\left(z'\right)} = \mathcal{L}\left(x, x'\right). \tag{51}$$

Here, the last equality arises from taking the ELBO in Eq. (15) and substituting the same deterministic encoders. Thus, with a deterministic encoder, the (marginal) log likelihood is equal to the ELBO.

## D  Experimental details

We generated 900 images in a single continuous video with a resolution of $256 \times 256$ pixels. The three balls had a diameter of 32 pixels. Between consecutive frames the balls moved by a full diameter in a random direction, as illustrated in Fig. 1A. The movement trajectory was picked by taking the previous trajectory and adding a uniform noise of $-2°$ to $+2°$. If the picked movement resulted in a collision, we sampled a new trajectory by doubling the noise range until a valid trajectory is found.

We trained the model in a classic self-supervised manner. We encoded one "base" frame, one "target" frame (the next frame in a video sequence), along with a number of random frames. As usual, the network was trained to distinguish between the target frame (adjacent to the base frame) and random frames. We then trained a linear decoder in a supervised manner to return the $(x, y)$ locations of the balls.

The encoder itself is a simple convolutional neural network, as shown in Fig. 2. It consists of 2 batch normalised convolutional layers with a kernel size of 3. The first layer uses ReLU as the activation function,

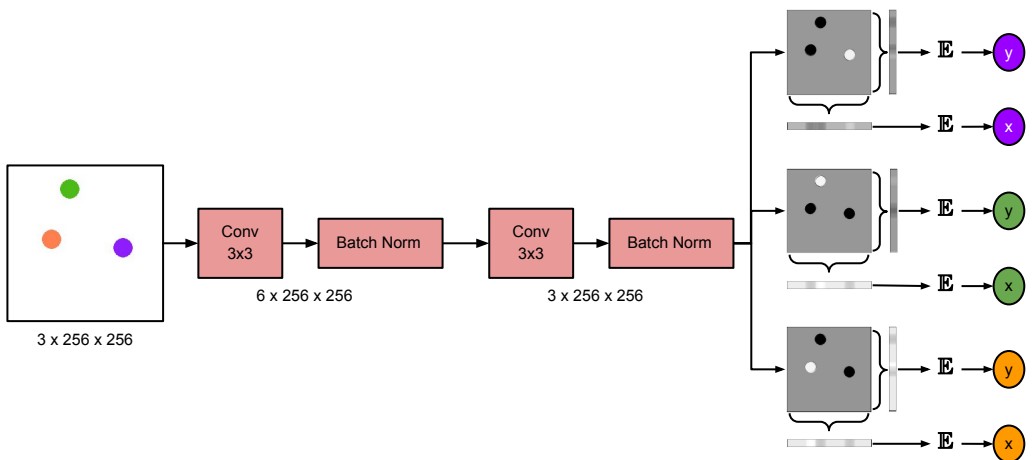

Figure 2: Architecture of the encoder neural network. The first of the two 3x3 convolutional layers outputs 6 feature maps and uses a ReLU activation. The second convolution outputs 3 feature maps and applies a sigmoid activation. For each of these 3 maps, we extract their centre of mass. This is done by summing each dimension and normalising it to 1. This is then used to perform a weighted average over the axis locations and get the final coordinates.

while the second layer uses a sigmoid. At the output of the convolutional layers, we have 3 feature maps, which we interpret as the locations of the 3 different balls. We finally extract these locations by computing the centre of mass of the feature maps, giving a vector of six numbers as output (the x and y locations of the centres of mass of each feature map). The training itself was performed by using stochastic gradient descent with a learning rate of 0.005 over the course of 30 epochs. The batches were made of 30 random pairs of consecutive frames. For any pair, we use the second frame as the positive example and we use the second frame of the other pairs in the batch, as the random negative examples, against which we contrast.

