# OpenReview forum: "InfoNCE is variational inference in a recognition parameterised model"
_TMLR — Accepted by TMLR_

### Review · Reviewer_JVxH · 2023-12-23

**Summary Of Contributions:**

This papers proposes a new understanding of the InfoNCE from the variational inference perspective, and claims its gain over the conventional mutual information understanding. To achieve this, they propose a recognition parameterized model that in turn, defines the generative probabilistic model as the bayes poster of the recognition model (i.e., the encoder). As a result, unlike VAEs, this model has no explicit decoder (can be a good thing), while they also show the variational lower bound (ELBO) is deeply connected to the mutual information (with a KL divergence accounting for the amortization gap). They also link the InfoNCE objective to the ELBO objective, which offers an alternative view. They show the benefit of this understanding using a rather toy experiment.

**Audience:**

Yes

**Claims And Evidence:**

No

**Requested Changes:**

The quality of this work could be significantly improved if they could propose new amendments to the manuscript (or show my misunderstandings):
1. Showing concrete new insights from the ELBO/RPM perspective.
2. Elaborate the connections and differences to the energy-based model formulation.
3. Elaborate the real difference between two perspectives.

Minor:

On page 2, there is an incomplete bracket "(see ...)"

**Strengths And Weaknesses:**

## Strengths
1. I would say that this paper is excellently written, starting from a intriguing and ambitious goal, and proceeding the arguments with convincing logic flows. It shows that the authors have plenty of knowledge of the literature as well as deep understandings of the essences of different approaches.

2. The paper has a good account of the related work. I fully understand that publishing paper can be quite uncertain in these days, and the authors set up good examples of how we could cope with situations like “because followup works got published earlier, is this work meaningless and hopeless now”. A not that up-to-date work can still give us lots of insights, and we should also account for its contributions in the context of its publishing date. Besides, given the elaborated descriptions of the publication history, to make a good evaluation of these arguments, I search for this paper and browse its submission history. I would say that by shifting the main theme from “InfoNCE is VAE” to “InfoNCE is “RPM”, the authors certainly have done a lot of polishing and it makes this work more understandable and less confusing.
3. I think that finding new perspective for understanding self-supervised objectives is a meaningful topic, and drawing connection to variational inference is certainly one of the most promising and interesting ways.
4. The recognition parameterised model (RPM) is an interesting probabilistic generative model, and it kind of inverts the usual convention from “decoder - variational encoder” to “encoder - variational decoder”.  In this way, RPM has an advantage over VAEs by discarding the heavy burden to explicitly reconstruct the outputs, which may help explains the gains of InfoNCE over VAEs for representation learning.

## Weakness

1. **Limited experiments and practical insights.** I would say that although the theory is interesting, the experiments are far from being inspiring or complete to show potential practical insights of this theory. Considering the toy experiment in Sec 5, it is actually a common choice to directly use the cosine similarity between $z$ and $z’$ for parameterizing $f_\theta(z,z’)$ in InfoNCE. And the new theory here does not add much new information.
2. **Connection to Poole et al. is not elaborated.** As far as I could see, the RPM model proposed has a close connection to the energy-based model discussed in Poole et al. (2019) which is prior to this work (and its preprint). Specifically, in their energy-based model (Eq. 3), they propose roughly the same formulation as Eq. 21b,  and they also established the connection between InfoNCE and a lower bound Eq.4 (the same lower bound as this paper Eq. 23) and the MI objective. Obviously, the energy-based model in Poole et al. is also a kind of decoder-free probabilistic model that bears great similarities to the definitions to this work (if not the same after addressing Problem 2), and they already show a variational inference of this model gives the InfoNCE loss. So with these similarities, their differences should be elaborated.
3. **Are MI and ELBO perspectives so different?** I note that the ELBO (as a lower bound of log likelihood) is *nothing but yet another  lower bound of MI*, since its upper bound, log likelihood, is also a variational lower bound of MI (up to constants). To see this, notice that

    $E_{P(x,y)} \log P(x,y) = E_{P(x,y)} \log Q(x,y) + KL(P(x,y), Q(x,y))\geq E_{P(x,y)} \log Q(x,y).$

    I think that this connection should be mentioned to give a holistic view of these different “lower bounds” and derivations. And in view of this limitation, this ELBO perspective is actually not much different from the MI lower bound perspective, e.g., in Poole et al.


Given these serious concerns, I think this work still lacks enough evidence to show that it is sufficiently different from prior works, or offer concrete new insights to algorithm design.

Ref:

Poole et al. On Variational Bounds of Mutual Information. In ICML. 2019.

---

> ### Comment · Editors_In_Chief · 2024-01-16
> **Response**
>
> Thanks for your positive and thoughtful review!
>
> **Experiments**
>
> We agree that the experiments here do not fully elaborate the promise of the approach.  That has been provided by the follow-up work discussed at length in the related work (e.g. Hasanzadeh et al. 2021, Wang et al. 2022, Walker et al. 2023a, Mollers et al. 2023, Wang et al. 2023).
> For instance, the RPM framework allows you to apply SSL-like approaches in traditional probabilistic models (Walker et al. 2023b), where the cosine similarity or unit-norm constraints really does not seem sensible, while, Jelly et al. (2023) used the additional flexibility of the RPM approach to develop contrastive methods for meta-learning in the few-shot setting.
>
> **MI vs VI**
>
> As discussed in Sec. 3.1 in the original manuscript, the standard VI objective --- the ELBO --- has a term that can be understood as a MI.
> In particular, the KL term in Eq. 2 can be understood as _minimizing_ MI between _data and latents_.
> InfoNCE can also be related to MI, but crucially, a different MI.
> In particular, InfoNCE _maximizes_ the MI between _different latents_.
> Thus, while both the ELBO and InfoNCE thus have a connection to MI, the particular MIs, and their effect on the objective and learning are very different.
> In particular, the traditional ELBO objective _minimizes_ the MI, while the traditional InfoNCE objective _maximizes_ the MI.
> This makes sense because they're different MIs.
> In particular, the traditional ELBO objective uses the MI between _data and latents_, while the InfoNCE objective uses the MI between _different latents_.
> Thus, even though VI and InfoNCE do both involve terms that can be interpreted as mutual informations, the exact mutual information and their effect on the objective is so different that it is not at all obvious whether any connection can thus be drawn between VI and InfoNCE (indeed, drawing the connection requires our very novel RPM framework).
>
> **Connection to Poole et al.**
>
> We have rewritten the abstract to clarify this point.
>
> In Poole et al., their goal is fundamentally different from this paper.
> Specifically, their goal is to develop bounds on the mutual information (indeed, their paper is entitled "On Variational Bounds of Mutual Information.")
> While the MI turns up in our paper, it is not the goal.
> Instead, the goal in our paper is to perform probabilistic inference.
> The only connection is that the MI happens to turn up in the resulting VI objective in some (very narrow) settings.
>
> Of course, Bayesian inference and information theory are radically different.
> In Bayesian inference, we are fundamentally solving a statistics problem.
> We have some prior beliefs about latent variables, and some data, and we want to update our beliefs about the latent variables conditioned on data.
> In contrast, when working with the MI, we are (at least historically) solving an engineering problem.
> For instance, we have a channel, which is often a physical system such as a wire, and we would like to transmit as much information as possible along that wire.
> The mutual information measures the information that can be transmitted.
>
> Of course, some confusion might arise from the use of the term "variational".
> The term "variational" is in widespread use, for instance in "variational calculus" and in the "variational method" in quantum mechanics (QM), both of which have nothing to do with either VI, or with variational bounds on mutual information (as in Poole et al. 2019).
> There is a common thread in these "variational" methods in that they bound some quantity; the bound has parameters, and you can optimize the parameters (the "variational" bit!) to tighten the bound.
> But this variational bounding strategy does not imply any kinship between the underlying problems of probabilistic inference, finding the ground state of a quantum system, or bounding the MI.
> So "variational" just implies a particular strategy to find a good bound.
> The underlying problem is really determined by what you're trying to bound: the model evidence/marginal likelihood in VI, the ground-state energy in QM or the MI in Poole et al. (2019).
>
> This is one of the key contributions of our work.
> In particular, we establish a new relationship between probabilistic inference and the mutual information between different latent variables.
> That enables us - in some circumstances - to use the bounds in Poole et al. (2019) to solve inference problems.
> Note however that this connection only holds in very restricted settings, such as when we use the recognition model as the approximate posterior and the prior is optimized (in Sec. 4.4).
> Importantly, outside of these settings, the exact connection to MI does not hold, and hence the results in Poole et al. (2019) cannot be applied.
> Thus the results in Poole et al. (2019) can only be applied to a very restricted subclass of RPMs.
>
> The original manuscript cited Poole et al. (2019).
>
> > On page 2, there is an incomplete bracket "(see ...)"
>
> Fixed.

---

> > ### Comment · Reviewer_JVxH · 2024-02-27
> >
> > Sorry for the late reply. I was just able to read this response today. I appreciate your detailed reply very much -- always an enjoyable read for me -- while I still have a few concerns that require further discussion. Particularly, I am not sure I understand this point (the key argument in your response that differentiates MI from ELBO):
> >
> > *"In particular, the traditional ELBO objective minimizes the MI, while the traditional InfoNCE objective maximizes the MI."*
> >
> > I don't think this claim is exactly the case. In Alemi et al, they still try to maximize the MI between $I(X;Z)$ leveraging bounds that are close to ELBO. I think it would be better to clarify these relationships. It would be unreasonable to minimize $I(X;Z)$ only, since one may just end up with some noise.

---

> ### Author Response · Authors · 2024-02-27
> **Response**
>
> I'm not sure about the paper you're referring to with "Alemi".  If you mean Poole et al. (2019) (as Alemi is a co-author on that paper), then I agree that Poole et al. are interested in maximising variational _bounds on the mutual information_.  But as discussed in my original response, variational _inference_ using the ELBO is completely different from variational _bounds on the mutual information_ considered by Poole et al. 2019.  Of course, they share the term "variational" which would seem to imply a connection.  But the term "variational" here is very generic and used in settings ranging including e.g. the variational calculus and the variational method in quantum mechanics, so does not imply a connection.
>
> The ELBO objective for VI can be loosely understood as a tradeoff: find a distribution Q(z|x) such that:
> * The expected log-likelihood remains high E_Q [P(z| x)].
> * The mutual information between z and x is small.
>
> That finds the broadest possible distribution over z such that most samples from that distribution still have high log-likelihoods.

---

### Review · Reviewer_dPq2 · 2024-01-04

**Summary Of Contributions:**

In the paper "InfoNCE is variational inference in a recognition parametrised model", the authors present a different view on the popular InfoNCE loss function, widely used in contrastive learning frameworks such as SimCLR. The authors argue that it is more helpful to see InfoNCE as an approximation to ELBO, rather than an approximation to MI (mutual information).

**Audience:**

Yes

**Claims And Evidence:**

Yes

**Requested Changes:**

MAJOR COMMENTS

* I must be confused, but I don't quite understand the meaning of \theta in Equation 8. The way I know InfoNCE, e.g. from SimCLR, the loss function simply uses exp(z^T z' / tau) terms, i.e. there is no matrix multiplication by matrix \theta, but instead there is a scalar temperature parameter \tau (which is typically held fixed). Please comment on why you need \theta matrix here, and whether one can also use parameter-less function f(z, z') that I gave above. I suggest to explicitly refer to SimCLR  here and explain how it fits into your notation/framework. It would also be useful to mention in this section that z and z' are typically normalized to have norm equal 1 (otherwise the objective does not make sense).

* This normalization issue becomes crucial in Section 5, which presents a toy experiment supposedly showing a failure mode of standard InfoNCE setup. However, this looks like a strawman to me, due to missing normalization. SimCLR uses cosine similarity, which can be written as z^T z' but ONLY IF z and z' are normalized to lie on the hypersphere and have norms 1. If they are not normalized, then z^T z' does not give cosine similarity (obviously) and it does not make sense to use that for what the authors call f_theta. What the authors show in Figure 1C as "linear" does NOT correspond to SimCLR because it uses unnormalized vectors in z^T z'.

  What about using exp(cos_similarity(z, z')) as f() and show that in Figure 1C instead of "linear"? Or at least in addition. Currently this whole section looks like  a strawman.


MINOR COMMENTS

* The paper contains many typos and could use a thorough proof reading. Page 2: "(see ...)". Page 2: "This seems to mirrors". Page 3: "Third,,".

* Section 4 title looks incomplete: "Recognition Parameterised Generative" -- this sounds like an adjective, not a noun. Is the word "model" missing?

**Strengths And Weaknesses:**

Disclaimer: I have not fully followed all the arguments, but overall the paper appears sensible. It is on topic for TMLR and may be of interest for its audience. My only substantial comment is on the experimental section; apart from that I only have very minor comments.

---

> ### Comment · Editors_In_Chief · 2024-01-16
> **Response**
>
> Thanks for your positive and careful review!
>
> > I must be confused, but I don't quite understand the meaning of \theta in Equation 8. The way I know InfoNCE, e.g. from SimCLR, the loss function simply uses $exp(z^T z' / \tau)$ terms, i.e. there is no matrix multiplication by matrix $\theta$, but instead there is a scalar temperature parameter $\tau$ (which is typically held fixed). Please comment on why you need $\theta$ matrix here, and whether one can also use parameter-less function $f(z, z')$ that I gave above. I suggest to explicitly refer to SimCLR here and explain how it fits into your notation/framework.
>
> If you look at the original paper introducing InfoNCE (i.e. Oord et al. 2018), they explicitly introduce $f$ with a matrix of parameters (see their Eq. 3).  In our notation, $\theta$ is that matrix of parameters.
>
> The connections between InfoNCE and SimCLR are well understood (the SimCLR paper states "This loss has been used in previous work (Sohn, 2016; Wu et al., 2018; Oord et al., 2018); for convenience, we term it NT-Xent (the normalized temperature-scaled cross entropy loss)."  We have added a note about this.
>
> > It would also be useful to mention in this section that z and z' are typically normalized to have norm equal 1 (otherwise the objective does not make sense).
>
> We have updated the text to mention this.
>
> > This normalization issue becomes crucial in Section 5
>
> Absolutely, the normalization issue is key.
> We absolutely agree that you probably could get a reasonable result here using the usual InfoNCE setup, with unit-norm latents.
> As such, the point here absolutely is not that you couldn't get something sensible in the usual setup.
> Instead, the point is that the real latent variables in this and other systems are not unit-norm.
> In particular, here, the real latent space is 6 dimensional, (x and y, for each of the 3 balls) and Euclidian, with no constraints normalization constraints.
> Thus, if you use the traditional SSL setup, with unit-norm latents, there is an inevitable mismatch between the real latent space, and the SSL latent space.
> One of the advantages of our approach was to enable the flexibility to use other non-unit-norm latent spaces that more closely mirror the real latent space.
> That is precisely what the experiment shows: that we can get sensible answers, despite dropping the unit-norm constraint, which thereby allows our latent space to more closely mirror the real latent space.
>
> We have added a note on this point to the Introduction, where we first discuss the experiment, and a more extensive discussion of this point to the experiment section.
>
> In the related work, there are a large number of follow-up papers that use the additional flexibility introduced by the RPM viewpoint in a number of areas (Hasanzadeh et al. 2021, Wang et al. 2022, Walker et al. 2023a, Mollers et al. 2023, Wang et al. 2023).
> For instance, the RPM framework allows you to apply SSL-like approaches in traditional probabilistic models (Walker et al. 2023b), where the unit-norm constraint really does not seem sensible, while, Jelly et al. 2023, used the additional flexibility of the approach develop contrastive methods for meta-learning in the few-shot setting.
>
> We have carefully proofread the paper, and fixed a number of typos including the ones raised directly:
>
> > Page 2: "(see ...)".
>
> Fixed.
>
> > Page 2: "This seems to mirrors".
>
> Fixed
>
> > Page 3: "Third,,".
>
> Fixed
>
> > Section 4 title looks incomplete: "Recognition Parameterised Generative" -- this sounds like an adjective, not a noun. Is the word "model" missing?
>
> Fixed.

---

> > ### Comment · Reviewer_dPq2 · 2024-02-29
> > **Thank you**
> >
> > Thank you for your reply. FYI the responses were invisible to the reviewers until a couple of days ago. So I could only read your response now.
> >
> > > If you look at the original paper introducing InfoNCE (i.e. Oord et al. 2018), they explicitly introduce with a matrix of parameters (see their Eq. 3). In our notation, theta is that matrix of parameters. The connections between InfoNCE and SimCLR are well understood [...] We have added a note about this.
> >
> > Thank you for clarification. I think the sentence that you added "Note that this loss has been used in downstream settings, such as SimCLR (Chen et al., 2020)" would be clearer if you add that in SimCLR theta is a scalar (inverse temperature).
> >
> > > Absolutely, the normalization issue is key. [...] We have added a note on this point to the Introduction, where we first discuss the experiment, and a more extensive discussion of this point to the experiment section.
> >
> > Thank you for inserting additional clarifications in the text. I think this is fine now.
> >
> > > For instance, the RPM framework allows you to apply SSL-like approaches in traditional probabilistic models (Walker et al. 2023b), where the unit-norm constraint really does not seem sensible, while, Jelly et al. 2023, used the additional flexibility of the approach develop contrastive methods for meta-learning in the few-shot setting.
> >
> > One other interesting connection may be Böhm et al ICLR 2023 (https://openreview.net/forum?id=nI2HmVA0hvt) who used a very similar $f()$  depending on $\|z-z'\|^2$ like your RBF in Eq 27. They used $1/(1+ \|z-z'\|^2)$ for the purpose of visualisation with two-dimensional $z\in\mathbb R^2$.

---

### Review · Reviewer_c21v · 2024-01-21

**Summary Of Contributions:**

This paper presents a novel perspective on self-supervised learning (SSL) by demonstrating that the popular InfoNCE objective is equivalent to Evidence Lower Bound (ELBO) in a new class of probabilistic generative models, dubbed Recognition Parametrized Models (RPMs). Some theoretical investigations are shown as follows:
1. The RPM ELBO can be written as the subtraction of mutual information with KL divergence between prior and data distribution of pairs, thus the optimal prior reduces to the mutual information up to a constant
2. Given the infinite sample, ELBO and InfoNCE are equivalent up to constant

**Audience:**

No

**Claims And Evidence:**

Yes

**Requested Changes:**

I believe this paper has a lot of room for improvement by clarifying the advantage of analyzing or implementing self-supervised learning with RPM; the author could do better on theory / experiments by showing some cases where RPM is better than other self-supervised objectives such as InfoNCE.

**Strengths And Weaknesses:**

## Strengths
1. The paper is clearly written with rigorous definitions and notations, providing a new theoretical concept in understanding self-supervised learning objectives.
2. In particular, the paper elaborates the equivalence between RPM ELBO and InfoNCE objective, and their relationship to mutual information, which is a common measure that previous works have focused on.

## Weakness
1. The major weakness of this paper would be the shortage of empirical validation. I suspect that the toy experiment demonstrates that the prior information is important in self-supervised learning, where it was nothing but simply using gaussian RBF for $f_\theta$. My question is, how does it corroborate your claim that RPM provides a new approach in self-supervised learning? Maybe providing more pragmatic experimental results on vision benchmarks (e.g., self-supervised learning on CIFAR-10, at its minimal approach) might be relevant to better deliver the theoretical finding.
2. In addition, although the concept of RPM is quite new to the literature, it seems like the equation and derivation is not new from previous literature. Consider $P_\theta(z,z’)$ as a family of energy-based model with energy $f_\theta(z,z’)$, then the proposed objective is nothing but Donsker-Varadhna (DV) objective [1], or MINE [2] that is derived for modern neural network architecture. While I agree that writing it into $P_\theta(z,z’)$ provides a more general formulation, I do not see much advantage in understanding SSL with such a perspective.

[1] Donsker, M. and Varadhan, S. Asymptotic evaluation of certain markov process expectations for large time, iv. Communications on Pure and Applied Mathematics, 36(2):183?212, 1983.

[2] Belghazi, Mohamed Ishmael, et al. "Mutual information neural estimation." International conference on machine learning. PMLR, 2018.

---

> ### Author Response · Authors · 2024-02-27
> **Response**
>
> Thanks for your positive and thoughtful review!
>
> > The major weakness of this paper would be the shortage of empirical validation. I suspect that the toy experiment demonstrates that the prior information is important in self-supervised learning, where it was nothing but simply using gaussian RBF for . My question is, how does it corroborate your claim that RPM provides a new approach in self-supervised learning? Maybe providing more pragmatic experimental results on vision benchmarks (e.g., self-supervised learning on CIFAR-10, at its minimal approach) might be relevant to better deliver the theoretical finding.
>
> This paper introduces a new family of Bayesian latent variable models, the RPM, and provides a novel and previously unsuspected theoretical link between VI in RPMs and pre-existing self-supervised learning methods.
>
> Follow up work discussed in the Related Work uses the additional flexibility introduced by RPMs in a number of areas (Hasanzadeh et al. 2021, Wang et al. 2022, Walker et al. 2023a, Mollers et al. 2023, Wang et al. 2023). For instance, the RPM framework allows you to apply SSL-like approaches in traditional probabilistic models (Walker et al. 2023b), while, Jelly et al. 2023, used the additional flexibility of the approach develop contrastive methods for meta-learning in the few-shot setting.
>
> > In addition, although the concept of RPM is quite new to the literature, it seems like the equation and derivation is not new from previous literature. Consider  as a family of energy-based model with energy , then the proposed objective is nothing but Donsker-Varadhna (DV) objective [1], or MINE [2] that is derived for modern neural network architecture. While I agree that writing it into  provides a more general formulation, I do not see much advantage in understanding SSL with such a perspective.
>
> We have rewritten the abstract to clarify this point.
>
> The goal of [2] is fundamentally different from this paper.
> Specifically, their goal is to develop an estimator for the mutual information (indeed, the paper is entitled ``Mutual information neural estimation.'')
> While the MI turns up in our paper, our goal is not to develop a new estimator of the mutual information.
> Instead, our goal is to perform probabilistic inference.  The only connection is that the MI happens to turn up in the resulting VI objective in some (very narrow) settings.
>
> Of course, Bayesian inference and information theory are radically different.
> In Bayesian inference, we are fundamentally solving a statistics problem.
> We have some prior beliefs about latent variables, and some data, and we want to update our beliefs about the latent variables conditioned on data.
> In contrast, when working with the MI, we are (at least historically) solving an engineering problem.
> For instance, we have a channel, which is often a physical system such as a wire, and we would like to transmit as much information as possible along that wire.
> The mutual information measures the information that can be transmitted.
>
> Our key contribution is to establish a new relationship between probabilistic inference and the mutual information between particular latent variables.
> That enables us --- in some circumstances --- to use the estimators of mutual information to solve inference problems.
> Note however that this connection only holds in very restricted settings, such as when the prior is optimized (in Sec. 4.4).
> Outside of these settings, the exact connection to MI does not hold, and hence plain mutual information estimators, such as those developed in [2] cannot be applied.
> Thus, VI in RPMs cannot always be reduced to [2], as it cannot always be reduced to maximizing a mutual information (only when you assume an optimal prior, as in Sec. 4.4).
>
> I was wondering whether you could elaborate more on the proposed link to [1]?  Notably, [1] was published in 1983, before VI emerged, let alone semi-supervised learning so it seems unlikely that it could provide a link between VI and semi-supervised learning.

---

### Decision · Action_Editor_V949 · 2024-03-21

**Recommendation:** Accept as is

**Comment:**

The paper introduces a novel perspective on the InfoNCE objective by framing it within the context of variational inference, specifically through the lens of Recognition Parametrized Models (RPMs). This approach diverges from the conventional understanding of mutual information in self-supervised learning, offering new conceptual framework that potentially enhances our comprehension of self-supervised learning objectives.

### Major Contributions:
- The paper's primary contribution is the innovative reinterpretation of the InfoNCE objective as an Evidence Lower Bound (ELBO) in the context of RPMs. This reinterpretation broadens the theoretical understanding of self-supervised learning.
- The paper is well-structured, presenting complex ideas in an accessible manner, thereby contributing to the theoretical discourse in the field.

### Points of Consideration:
- Multiple reviewers highlight the need for more empirical evidence to support the theoretical claims made in the paper. While the theoretical framework is robust, additional experiments, particularly on more practical or larger-scale datasets, would bolster the paper's impact and relevance.

### Recommendation:
Despite the concerns regarding empirical validation and the need for a more distinct differentiation from existing literature, the paper presents a significant theoretical contribution that enriches our understanding of self-supervised learning. Therefore, I recommend the acceptance of this paper.

**Audience:**

The paper is relevant to various fields of study including Bayesian inference, self-supervised learning, and contrastive learning.

**Claims And Evidence:**

The paper presents a commendable level of rigor in exploration of the InfoNCE objective from a variational inference perspective. In particular, the authors articulate a compelling argument, underpinned by a theoretical framework that aligns the InfoNCE objective with the Evidence Lower Bound (ELBO) within the novel context of Recognition Parametrized Models (RPMs).